# Proinflammatory cytokines and ARDS pulmonary edema fluid induce CD40 on human mesenchymal stromal cells—A potential mechanism for immune modulation

Erin M. Wilfong[1¤]*, Roxanne Croze[2], Xiaohui Fang[2], Matthew Schwede[1], Erene Niemi[3], Giselle Y. López[4], Jae-Woo Lee[5], Mary C. Nakamura[1,3], Michael A. Matthay[6]

1 Department of Medicine, University of California San Francisco, San Francisco, California, United States of America, 2 Cardiovascular Research Institute, University of California San Francisco, San Francisco, California, United States of America, 3 San Francisco Veterans Affairs Medical Center, San Francisco, California, United States of America, 4 Department of Pathology, University of California San Francisco, San Francisco, California, United States of America, 5 Department of Anesthesia, University of California San Francisco, San Francisco, California, United States of America, 6 Departments of Medicine and Anesthesiology, Cardiovascular Research Institute, University of California San Francisco, San Francisco, California, United States of America

¤ Current address: Department of Medicine, Vanderbilt University Medical Center, Nashville, Tennessee, United States of America

* erin.m.wilfong@vumc.org

## Abstract

Human mesenchymal stem/stromal cells (hMSCs) are a promising therapy for acute respiratory distress syndrome (ARDS) and other inflammatory conditions. While considerable research has focused on paracrine effects and mitochondrial transfer that improve lung fluid balance, hMSCs are well known to have immunomodulatory properties as well. Some of these immunomodulatory properties have been related to previously reported paracrine effectors such as indoleamine-2,3-dioxygenase (IDO), but these effects cannot fully account for cell-contact dependent immunomodulation. Here, we report that CD40 is upregulated on hMSCs under the same conditions previously reported to induce IDO. Further, CD40 transcription is also upregulated on hMSCs by ARDS pulmonary edema fluid but not by hydrostatic pulmonary edema fluid. Transcription of CD40, as well as paracrine effectors TSG6 and PTGS2 remained significantly upregulated for at least 12 hours after withdrawal of cytokine stimulation. Finally, induction of this immune phenotype altered the transdifferentiation of hMSCs, one of their hallmark properties. CD40 may play an important role in the immunomodulatory effects of hMSCs in ARDS and inflammation.

## Introduction

Human mesenchymal stem/stromal cells (hMSCs) have emerged as a promising therapy for myriad inflammatory conditions including graft versus host disease, systemic lupus

**Data Availability Statement:** All relevant data are within the manuscript and its Supporting Information files.

**Funding:** This work was funding by the National Institutes of Health (R01HL134828 MAM, T32AR007304 EMW, AG046282 MCN) www.nih. gov. MCN is also supported by a VA Merit Review (www.va.gov) and the Russell/Engleman Rheumatology Research Center (https:// rheumatology.ucsf.edu/russell-engleman). The funders had no role in study design, data collection and analysis, decision to publish, or preparation of the manuscript.

**Competing interests:** The authors have declared that no competing interests exist.

erythematosus, multiple sclerosis, sepsis, and acute respiratory distress syndrome (ARDS) [1, 2]. Our research group has focused on the potential role of hMSCs for treatment of the ARDS [3].

ARDS represents a state of immune activation in which inflammatory insults such as sepsis, transfusions and major trauma lead to increased activation of the innate immune system— particularly macrophages and neutrophils. Alveolar macrophage activation leads to secretion of inflammatory cytokines, including interleukin-1β (IL-1β), tumor necrosis factor (TNF), interleukin-6 (IL-6), and interleukin-8 (IL-8), which lead to additional neutrophil activation {Chollet-Martin, 1996 #139; Strieter, 1993 #134}. This combined activation of neutrophils and macrophages mediates some of the tissue injury in ARDS [4, 5]. While aberrations in innate immunity have been well described, the humoral immune system is also dysregulated. Activation of CD4+ T cells has also been reported in a murine model of lipopolysaccharide (LPS)- induced ARDS [6]; T regulatory ($T_{reg}$) cells are also involved in resolution of acute lung injury in mice and humans [7].

hMSCs have many beneficial immunomodulatory effects. Through soluble mediators such as indoleamine-2,3-dioxygenase (IDO) [8] and prostaglandin E2 (PGE2) [9], hMSCs promote the M1 to M2 transition of activated macrophages. hMSCs also promote T cell suppression through the IL-1β mediated secretion of transforming growth factor-β (TGF-β) [10]. Despite the well-characterized paracrine effects, cell-cell contact is also required for induction of $T_{reg}$ cells [11] and some aspects of macrophage modulation [10].

Previously, our laboratory has demonstrated that a mixture of inflammatory cytokines, termed CytoMix (50 ng/mL TNF-α, interferon (IFN)-γ, and IL-1β) recapitulated the effects of ARDS pulmonary edema fluid on type II alveolar cells, including loss of tight junctions and impaired fluid clearance [12]. Re-analysis of prior microarray data revealed that CytoMix also induced hMSCs transcription of CD40 [13]. Here, we report the conditions under which CD40 is transcribed and importantly also report that CD40 upregulation develops after exposure of hMSCs to human ARDS pulmonary edema fluid.

## Materials and methods

### Mesenchymal stem/stromal cells

hMSCs were obtained from Dr. D.J. Prockop at the Institute for Regenerative Medicine in Texas A&M Health Science Center (IRM), Dr. Shibani Pati at the Blood Systems Research Institute (BSRI), and Dr. David McKenna at the University of Minnesota (UM). Cell lines were validated using qPCR for the presence of CD73, CD90, and CD105 expression, as well as the absence of CD11b, CD14, CD34, CD45, CD19, CD79A, CD54, and HLADRB expression (S1 Fig). Cells were cultured as previously described [14].

### hMSC treatment with cytokines or LPS

For these experiments, hMSCs were plated at $1.6 \times 10^5$ cells/cm$^2$ on 60mm untreated tissue culture plates in DMEM-F12 supplemented with antibiotics. TNF-α, IL-1β, and IFN-γ were obtained from R&D Systems. As previously described, CytoMix consisted of 50 ng/mL TNF-α, IFN-γ, and IL-1β [12]. For cytokine array experiments, TNF-α, IFN-γ, and IL-1β were used at 50 ng/mL in various combinations. LPS concentrations ranged from 5 pg/mL– 1 mg/mL (List Labs or Sigma). In all cases, the hMSCs were exposed to the experimental condition for 24 hours prior to harvesting, unless otherwise noted.

## Human pulmonary edema fluid

Human pulmonary edema fluid was previously collected by our laboratory immediately after endotracheal intubation of patients with ARDS secondary to sepsis or hydrostatic pulmonary edema, with Institutional Review Board approval at the University of California, San Francisco [15]. Samples were frozen at -80˚C until use in these studies. Undiluted pulmonary edema fluid from four donors with either ARDS or hydrostatic edema fluid was thawed, pooled, and centrifuged at 10,000 rpm for 2 minutes to remove cellular debris. The gender of all hMSC lines was determined by XIST levels; two male and two female lines were utilized to avoid gender confounding. hMSCs were plated at $1.5 \times 10^4$ cells/cm$^2$ for 12 hours, washed with PBS and incubated with undiluted pulmonary edema fluid for 24 hours. RNA was isolated as described below.

## Real Time PCR (RT-PCR) analysis

Total RNA from hMSCs was extracted using Qiagen® QiaShredder and either RNeasy mini or RNAeasy plus mini kits according to the manufacturer's methods. RNA was quantified using a ND-1000 (ThermoFisher) and 260/280 absorbance; 1 μg RNA was converted to cDNA using the cDNA iScript synthesis kit (Biorad). RT-PCR was performed in technical triplicate on each biologic replicate using the Agilent platform, StepOnePlus System (Applied Biosystems) and Fast Sybr® Green Mastermix (Applied Biosystems); primers are shown in Table 1. Run cycle: PCR was activated at 95˚C for 20 seconds followed by 40 cycles of 3 seconds at 95˚C followed by 20 seconds at 60˚C. All RT-PCR analyses for CytoMix and cytokine array were carried out on at least 5 biologic replicates; only four biologic replicates were conducted for the human pulmonary edema fluid samples due to limited reagents. Lipopolysaccharide (LPS) experiments were only conducted in biologic triplicate due to futility. $\Delta C_T$ values were normalized to housekeepers TBP and EIF2E2.

## hMSC differentiation

hMSC differentiation studies were conducted using the R&D Systems human mesenchymal stem cell functional identification kit according to the manufacturer's directions using fatty acid binding protein 4 (FABP4), osteocalcin, and aggrecan as markers of adipocyte, osteoblast, and chondrocyte differentiation, respectively. Adipogenesis was further evaluated using oil red O staining [16]. Images were transmitted to a pathologist in blinded manner. Positive and negative cells were quantified in Image J; a minimum of 99 cells were counted per cohort.

## Flow cytometry

Four-color flow cytometry was performed on a FACS-Calibur II instrument. KG-1 cells differentiated to dendritic cells with PMA and ionomycin [17] were used as positive controls for HLA-DR (ThermoFischer, MEM-12 clone) and CD40 (Novusbio, 5C3 clone). Two representative cell lines were harvested by incubation with Accutase (Gibco) for five minutes. Analyses were conducted with FlowJo version 10. Cells were gated on hMSCs by forward and side scatter. The CD40 positive gate was determined *a priori* on the unstained sample and then applied to the stained control and CytoMix-exposed cells.

## Statistical analysis

Statistical analyses for all RT-PCR experiments were conducted in GraphPad Prism (V8.2.1). For all direct comparisons between experimental groups using cytokine mixtures (n = 6), Mann-Whitney U tests were utilized. Pulmonary edema fluid experiments (n = 4) were

**Table 1. RT-PCR primers.**

| Target | | Primer | #BP |
|---|---|---|---|
| CD73 | F | GCCGCTTTAGAGAATGCAAC | 116 |
| | R | TTTCATCCGTGTGTCTCAGG | |
| CD90 | F | GGACTGAGATCCCAGAACCA | 95 |
| | R | TTAGGCTGGTCACCTTCTGC | |
| CD105 | F | GCACATCCTGAGGGTCCTG | 102 |
| | R | ATGAGGACGGCATCGAGA | |
| CD11b | F | TCTACCAGTGCGACTACAGCA | 75 |
| | R | ACATGTTCACGGCCTCCAC | |
| CD14 | F | AAGCACTTCCAGAGCCTGTC | 82 |
| | R | CAGCAGCAACAAGCAGGAC | |
| CD34 | F | CACCCTGTGTCTCAACATGG | 115 |
| | R | GGACAGAAGAGTTTGTGTTTCCA | |
| CD45 | F | AGGAAATTGTTCCTCGTCTGA | 76 |
| | R | GAAGTCAGCCGTGTCCCTAA | |
| CD19 | F | TGGTCCTGAGGAGGAAAAGA | 103 |
| | R | ACGTTCCCGTACTGGTTCTG | |
| CD79A | F | GGGGATCATCCTCCTGTTCT | 79 |
| | R | GAGCTTCTCGTTCTGCCATC | |
| CD54 | F | TGCTATTCAAACTGCCCTGA | 80 |
| | R | AGTTCCACCCGTTCTGGAGT | |
| HLA-DRB1 | F | TCTGCATTTCAGCTCAGGAA | 79 |
| | R | GCCAACATAGCTGTGGACAA | |
| HLA-DRA1 | F | GTTGGGCTCTCTCAGTTCCA | 121 |
| | R | TTGGCTTTCCTGCTGAGTCT | |
| HLA-DPA1 | F | CCCTGTTGGTCTATGCGTCT | 96 |
| | R | CCCTGTGGAGGTGAAGACAT | |
| HLA-DQA1 | F | CAGAGGGACCGTAAAACTGG | 79 |
| | R | TCTGCATTTCAGCTCAGGAA | |
| CD68 | F | TTCCCCTATGGACACCTCAG | 86 |
| | R | TTGTACTCCACCGCCATGTA | |
| CD40 | F | GTGTCCTGCACCGCTCAT | 78 |
| | R | GCTCGCAGATGGTATCAGAA | |
| CD83 | F | GGTGGTGAAGAGAGGATGGA | 81 |
| | R | AGAACCATTTTGCCCCTTCT | |
| CD80 | F | TTGTTCTGAAGTATGAAAAAGACG | 84 |
| | R | GGTGTAGGGAAGTCAGCTTTG | |
| CD86 | F | GCCCAGAATTCTAAGCTGGT | 76 |
| | R | CCACCCAGACTGAGGAGGTA | |
| TSG6 | F | GCTAGAGGCAGCCAGAAAAA | 86 |
| | R | GCTTCACAATGGGGTATCCA | |
| PTGS2 | F | CTAGAGCCCTTCCTCCTGTG | 78 |
| | R | TTGAATCAGGAAGCTGCTTTT | |
| IL1RA | F | TTGCAAGGACCAAATGTCAA | 79 |
| | R | GGATTCCCAAGAACAGAGCA | |

analyzed using unpaired two-tailed *t* test due to limited pulmonary edema fluid sample and reduced power. If multiple comparisons were performed, a Kruskal-Wallis ANOVA (KW) was performed first to determine if a difference was present across all groups and, if significant, Mann-Whitney U-tests were performed to determine which groups were statistically different. For flow cytometry, the geometric mean and standard deviation of positive populations and MFI are reported (FlowJo V10). Statistical comparisons between flow cytometry population frequencies were made using an unpaired *t* test (GraphPad Prism V8.2.1). Fisher's exact tests were utilized to compare positive cell frequency for quantifications (https://www.graphpad.com/quickcalcs/contingency2/). P values < 0.05 were considered statistically significant. S1 Appendix includes raw data and additional statistical analyses in accordance to the minimal data set definition.

## Results

### CytoMix induces transcription of CD40, CD83, and HLA-DR in hMSCs

Prior microarray results indicated that the hMSCs differentiated into a dendritic cell-like phenotype with CD40, CD83, CD68, and MHC class II expression when exposed to CytoMix, but these studies were done with a single biologic replicate [13, 18]. To confirm these findings at a transcriptional level, hMSC were exposed to CytoMix. RT-PCR revealed markedly increased transcription of CD40, CD83, and HLA-DR (Fig 1). Additional studies of co-stimulatory molecules CD80, CD86, CD40L, and CD163 revealed no increased transcription. The transcription of CD40, CD83, and HLA-DR were not induced by any concentration of LPS (S2 Fig). Thus, the induction of these immune markers appeared to be due to stimulation with specific cytokine(s) rather than activation in general.

### CD40 transcription requires the same cytokine stimulation as IDO upregulation

Ren et al. previously reported that hMSCs exposed to IFN-γ plus TNF-α or IL-1β express IDO, and these conditions were associated with both paracrine and contact-mediated effects on T cells [19]. Thus, we carried out a cytokine array to determine the requisite cytokine exposure for induction of CD40, CD83 and MHC-class II transcription. hMSCs were exposed for 24 hours to a single cytokine (either TNF-α, IL1-β, or IFN-γ) or combinations of these three cytokines (Fig 1). Like IDO, CD40 was also strongly induced by IFN-γ plus one additional cytokine, although a small increase in transcription was noted for other conditions. CD83 was strongly induced by TNF-α + IFN-γ, but the increase induced by IL-1β + IFN-γ was not significant (p = 0.056). As expected, based on prior reports [20], transcription of HLA-DR was induced by IFN-γ alone. Paracrine effector molecule TSG-6 was expressed under similar conditions as IDO and CD40; PTGS2 was upregulated under all conditions except TNF-α monostimulation.

### CD40 is expressed on the cell surface after CytoMix exposure

While the changes in mRNA expression for these immune markers are substantial, RNA transcription does not necessarily correspond to increased protein levels and transport to the cell surface. Flow cytometry demonstrated CD40 cell surface expression in a significant subset of cells (Fig 2A and 2B). While the MFI shift was not statistically significant, there was a trend towards increased MFI in the CytoMix exposed samples. (Fig 2C) In contrast, no surface expression of CD83 nor properly assembled HLA-DR with the MEM-12 anti-HLA-DR clone was demonstrated despite significant gene transcription. HLA-DR upregulation by

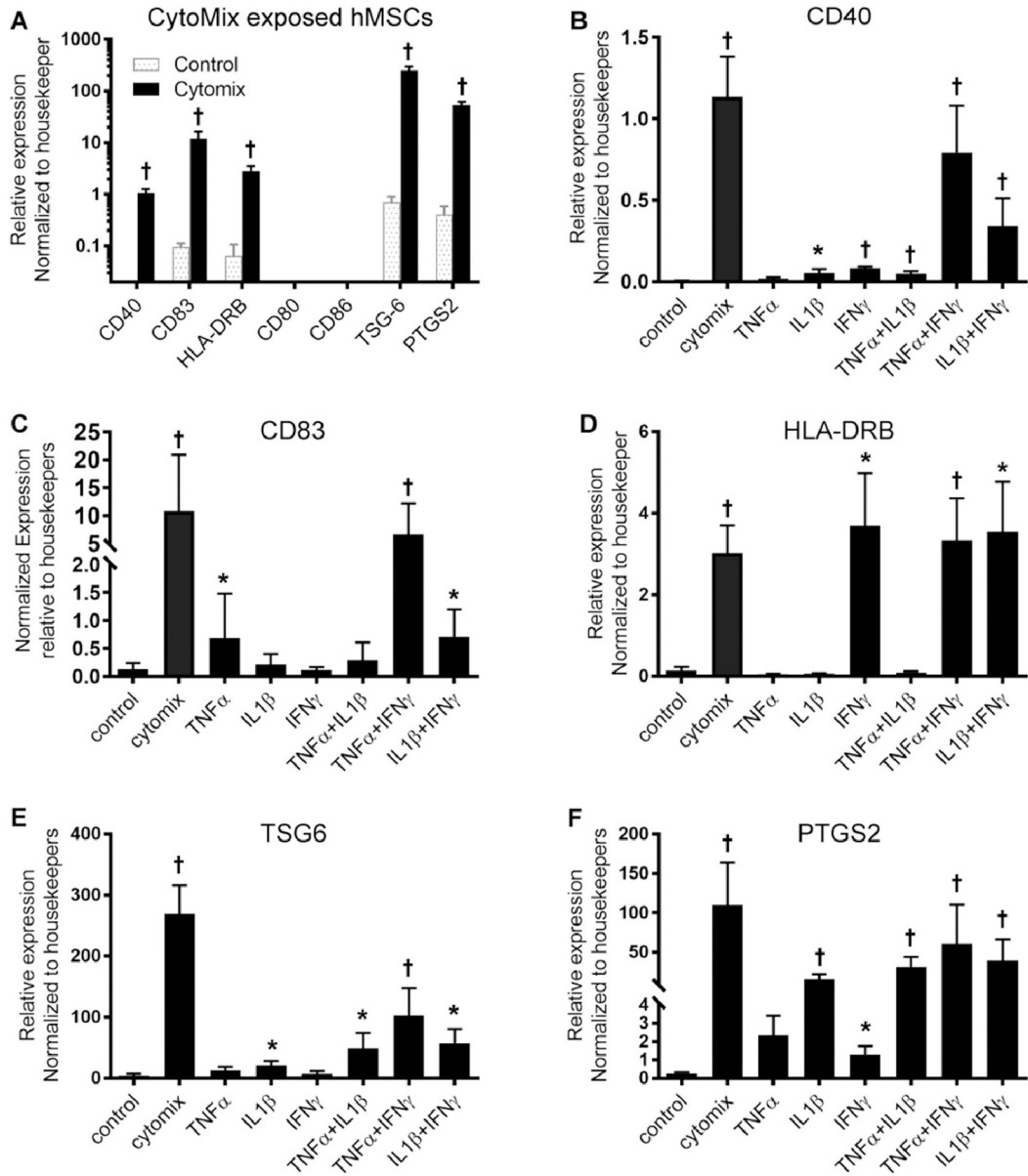

**Fig 1. Quantitative RT-PCR results.** (A) Transcription of highlighted genes for hMSCs with and without CytoMix exposure. Cytokine arrays to elucidate the cytokine milieu required for transcription of (B) CD40 (KW p<0.001), (C) CD83 (KW p<0.001), (D) HLA-DRB (KW p = 0.003), (E) TSG-6 (KW p<0.001), and (F) PTGS2 (KW p<0.001) are also shown. mRNA expression levels were normalized to housekeeper genes EIF2E2 and TBPData shown as mean ± SEM, and *p < 0.05, †p < 0.01 compared to control using Mann-Whitney U-test.

transcriptional analysis has been observed in prior reports but was not previously correlated with analysis of surface expression [21].

## CD40, TSG-6, and PTGS2 transcription is induced by ARDS pulmonary edema fluid

While CD40 could be reliably transcribed and expressed in an in vitro system of ARDS, using stimulation with a mix of inflammatory cytokines, we sought to determine if more in vivo-like

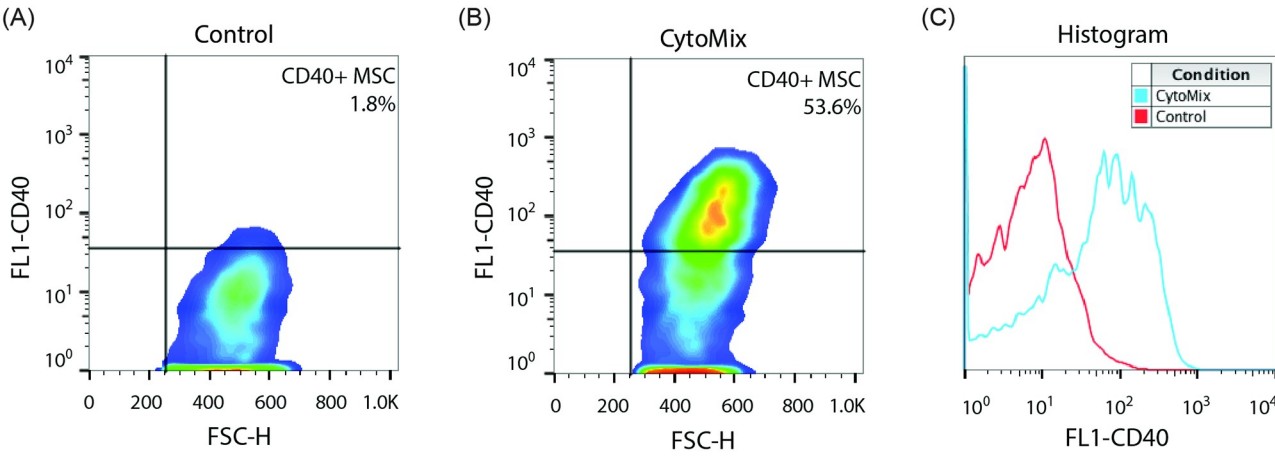

**Fig 2. Flow cytometry demonstrates cell surface expression of CD40 before (A) and after (B) exposure to CytoMix.** The CD40+ population increased from 3.2 ± 1.9% to 52.7 ± 7.4%, (p = 0.02). A representative histogram shows an increase in MFI from 2.3 ± 14.8 to 26.5 ± 118 after CytoMix stimulation is shown in panel C. Data reported as geometric mean ± SD, p values were determined using unpaired t tests.

conditions might be similar. hMSCs were exposed to undiluted pulmonary edema fluid from patients with either ARDS or hydrostatic pulmonary edema (e.g. heart failure). Exposure to either pulmonary edema fluid failed to induce CD83 or HLA-DRB transcription. CD40 transcription was not altered by exposure to hydrostatic pulmonary edema fluid; however, exposure to ARDS pulmonary edema fluid resulted in a 5-fold increase of CD40 mRNA levels in hMSCs (p-value 0.04). Confirmatory flow cytometry to examine CD40 protein expression could not be performed due to the small quantity of pulmonary edema fluid available. The paracrine molecules were similarly investigated. PTGS2 had increased transcription for both hydrostatic and ARDS pulmonary edema fluid (~20-fold and 1000-fold increase, p values 0.002 and 0.02, respectively), but TSG-6 transcription was induced solely by ARDS pulmonary edema fluid (6-fold increase, p value 0.02).

### Induction of the immunologic phenotype affects pluripotency

Pluripotency is a key feature of mesenchymal stromal cells, and hMSCs are capable of differentiating into osteoblasts, adipocytes, or chondrocytes, and transdifferentiation is possible by changing the culture conditions [22]. Given the robust transcriptional changes induced by CytoMix, we questioned if CytoMix exposure would also alter hMSC pluripotency. Three representative cell lines were differentiated into chondrocytes, adipose tissue, and osteoblasts with or without a 24 hour pre-treatment with CytoMix using standard conditions. Successful differentiation was assessed by immunofluorescence for FABP4 (adipogenesis), osteocalcin (osteogenesis), and aggrecan (chondrogenesis). While the percent of aggrecan positive chondrocytes (7.1% v. 8.0%) and osteocalcin positive osteoblasts (87.0% v.83.8%) were unchanged after CytoMix exposure, CytoMix reduced FABP4 positivity from 35.4% to 5.5%, which indicates inhibition of adipogenesis. This finding was confirmed with oil red O staining by light microscopy, which also demonstrated a reduction in oil red O positive cells after CytoMix exposure (70.9% v. 16.7%) (Fig 3). Thus, preconditioning with CytoMix appears to result in reduction of pluripotency, albeit in an in vitro setting.

### Discussion

The main findings of our studies can be summarized as follows. The pro-inflammatory mixture CytoMix induces CD40 cell surface expression on hMSCs under the same conditions

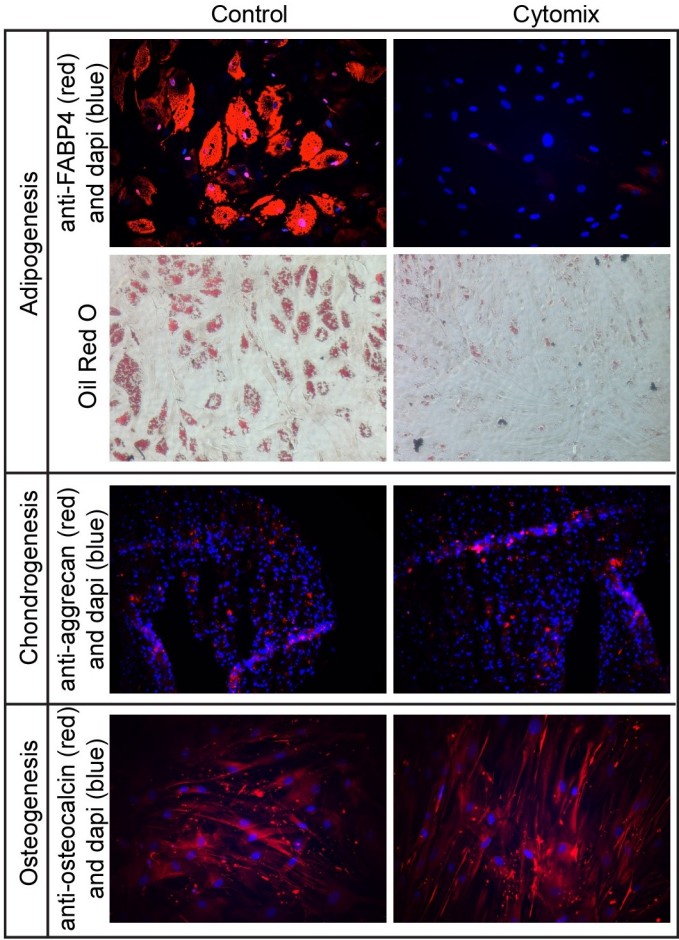

**Fig 3. Pleuripotency of hMSCs before and after CytoMix exposure.** hMSCs were treated with 24 h DMEM-F12 medium or DMEM-F12 medium + CytoMix prior to exposure to standard differentiation conditions. 24 hour pre-treatment with CytoMix prevented adipogenesis as demonstrated by the lack of FABP4 staining by immunofluorescence (35.4% v. 5.5% positivity) and oil red O staining by light microscopy (70.9% v. 16.7% positivity). Differentiation of osteoblasts (87.0% v. 83.8%) or chondrocytes (7.1% v.8.0%) was unaffected.

required for IDO expression, which is associated with a potent immunomodulatory phenotype with both paracrine and cell contact-mediated effects. Pulmonary edema fluid from ARDS patients, but not from hydrostatic edema fluid, also led to upregulation of CD40 gene expression. Furthermore, CytoMix upregulated the expression of several anti-inflammatory cytokines and paracrine effectors, which was also demonstrated with ARDS pulmonary edema fluid. Finally, CytoMix exposure decreased in vitro pleuripotency of hMSCs and inhibited hMSCs differentiation into adipocytes.

Previous studies of hMSCs have documented that they do not express standard immune markers such as MHC-class II, CD40, CD80/86, and others [23], although there is one mention of CD40 upregulation observed in hMSCs after IFN-γ exposure [24]. In our studies, small quantities of CD40 transcription were detected with IFN-γ alone, but CD40 transcription increased markedly when IFN-γ was combined with IL-1β or TNF-α. Ren et al. reported that hMSCs exposed to IFN-γ plus either IL-1β or TNF-α develop a potent immunomodulatory phenotype that partially relies on close cell-cell proximity [19]. While that study attributed the

immunomodulation to IDO, we postulate that the expression of CD40 may contribute to this immunomodulatory phenotype.

CD40 is a key molecule in the immune system, and CD40+ hMSCs likely have a multi-modal approach to modulating the inflammatory response of ARDS. Given the impressive pre-clinical data using hMSC treatment for ARDS [25, 26] and early clinical studies in systemic lupus erythematosus [27], it is clear that hMSC treatments can have significant immunomodulatory effects. Prior literature lends credence to the possibility of T cell modulation by CD40+ hMSCs. Interestingly, CD40L+/CD14+ peripherally circulating monocytes and CD40L+ T cells [28] have been reported in active systemic lupus erythematosus. While the majority of research efforts have focused on CD40 co-stimulation of B cells, CD40-CD40L binding also causes signaling in the CD40L expressing cell [29]. CD40L is important in the negative selection of auto-reactive thymocytes [30]. In CD4+ T cells, CD40-CD40L binding leads to increased level of IL-10 and IFN-γ [31]; in the presence of IFN-γ, CD40L binding can also lead to the generation of nitric oxide [32]. Additionally, low levels of CD40 expression on dendritic cells leads to marked expansion of $T_{reg}$ cells in a murine model of *Leishmania donovani* infection [33], and hMSCs promote the development of $T_{reg}$ cells *in vitro* [34]. Thus, CD40$^+$ hMSCs may simultaneously decrease proliferation of activated T cells while promoting the expression of $T_{reg}$ cells.

Beyond direct CD40 mediated modulation of the T cell repertoire, hMSCs expressing CD40 could serve as a decoy receptor for soluble CD40L, which been implicated in patients with ARDS. Soluble CD40L has been implicated in the development of transfusion associated acute lung injury [35], sickle cell-related acute chest syndrome [36], and sepsis [37]. Once hMSCs lodge within the pulmonary microcirculation following intravenous administration, local reductions of CD40L could ameliorate endothelial dysfunction and possibly reduce inflammatory pulmonary edema.

A significant challenge in understanding hMSC biology is their significant pluripotency and differential responses to various conditions. In this study, the transcriptional response of hMSCs to clinically relevant stimuli (CytoMix or pulmonary edema fluid) was variable. Even for ARDS, a well characterized condition, our group has reported sub-phenotypes characterized by varying levels of inflammatory cytokines, which correlate with clinical course and mortality [38]. Thus, the in vivo behavior of hMSCs in individual patients may be variable and could represent a challenge in clinical trials.

One strength of this study is the robust and reproducible response of the hMSCs from multiple sources to the CytoMix stimuli to produce this potentially immunomodulatory phenotype. There was significant concern that this phenotype, although reproducible in technical replicates in a microarray, might not be reproducible among multiple human donors or might be a result of *ex vivo* expansion and manipulation of the hMSCs. For this reason, six human donors were used for the cytokine-exposed transcriptional studies from three independent sources; the gender ratio was four females to two males. Given the robust nature of this phenotype under specific conditions and the potential variability between patients and human disease states, preconditioning with CytoMix to induce an immunomodulatory phenotype may be useful to ensure a uniform hMSC functional phenotype in clinical trials, but determining the importance of this effect will require dditional *in* vivo experiemnts.

A limitation of our study is that we do not have direct evidence that CD40 is upregulated on hMSCs in patients treated with hMSCs for ARDS. One potential future avenue of research is to retrieve hMSCs from the bronchoalveolar lavage of patients with the ARDS, which is planned for subset of patients of an ongoing randomized clinical trial of hMSCs for ARDS (NCT03818854). We believe, however, that the upregulation of gene transcription of CD40 upon exposure to pulmonary edema fluid provides strong circumstantial evidence that CD40

may be relevant in ARDS patients receiving hMSCs. Future studies will need to more directly examine how hMSC CD40 expression contributes to the suppression of T cell and macrophage hyperinflammatory states.

## Conclusions

A combination of inflammatory cytokines in the clinically relevant condition ARDS leads to upregulation of CD40 gene transcription and cell surface expression on bone marrow-derived mesenchymal stem/stromal cells. Expression of CD40 provides hMSCs with a new pathway to interact with other immune cells. CD40 expression is induced under the same cytokine conditions as IDO, which has previously been shown to be a potent immunomodulatory phenotype of hMSCs.

## Supporting information

**S1 Fig. hMSC validation by quantitative RT-PCR.** (A) All hMSC cell lines expressed CD73, CD90, and CD105 and (B) did not express high levels of CD11, CD14, CD34, CD45, CD19, CD79A, CD54, CD40, or HLA-DRB. mRNA expression levels were normalized to housekeeper genes EIF2E2 and TBP.
(TIF)

**S2 Fig. CD40, CD83, and HLA-DR expression is not induced by LPS.** Quantitative RT-PCR failed to reveal any increased transcription of CD40, CD83, or HLA-DR after 24 hours of exposure to varying concentrations of LPS. mRNA expression levels were normalized to housekeeper genes EIF2E2 and TBP.
(TIF)

**S1 Appendix. Underlying data and descriptive statistics for all figures and experiments in accordance with PlosONE data transparency standards.**
(DOCX)

## Author Contributions

**Conceptualization:** Erin M. Wilfong, Roxanne Croze, Xiaohui Fang, Matthew Schwede, Jae-Woo Lee, Michael A. Matthay.

**Data curation:** Erin M. Wilfong, Erene Niemi, Giselle Y. López.

**Formal analysis:** Erin M. Wilfong, Roxanne Croze, Matthew Schwede.

**Funding acquisition:** Mary C. Nakamura, Michael A. Matthay.

**Investigation:** Erin M. Wilfong, Erene Niemi.

**Methodology:** Erin M. Wilfong, Roxanne Croze, Xiaohui Fang, Matthew Schwede, Erene Niemi, Giselle Y. López, Mary C. Nakamura, Michael A. Matthay.

**Project administration:** Michael A. Matthay.

**Resources:** Giselle Y. López, Jae-Woo Lee, Mary C. Nakamura, Michael A. Matthay.

**Supervision:** Xiaohui Fang, Erene Niemi, Mary C. Nakamura, Michael A. Matthay.

**Visualization:** Mary C. Nakamura.

**Writing – original draft:** Erin M. Wilfong.

**Writing – review & editing:** Erin M. Wilfong, Roxanne Croze, Xiaohui Fang, Matthew
Schwede, Erene Niemi, Giselle Y. López, Jae-Woo Lee, Mary C. Nakamura, Michael A.
Matthay.

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
