## [Decision Letter · Decision Letter 0]

28 Jun 2020

PONE-D-20-17990

Proinflammatory cytokines and ARDS pulmonary edema fluid induce CD40 on human mesenchymal stromal cells – a mechanism for immune modulation

PLOS ONE

Dear Drs. Wilfong and Matthay,

Thank you for submitting your manuscript to PLOS ONE. After careful consideration, we feel that it has merit but does not fully meet PLOS ONE’s publication criteria as it currently stands. Therefore, we invite you to submit a revised version of the manuscript that addresses the points raised during the review process.

Please provide quantification of Fig 2 and Fig 3. Complete the the statistical analysis section and clarify some of the comments raised by reviewer 1. You don't have to provide new data. 

We look forward to receiving your revised manuscript.

Kind regards,

You-Yang Zhao

Academic Editor

PLOS ONE

Journal Requirements:

Additional Editor Comments (if provided):

Reviewers' comments:

Reviewer's Responses to Questions

**Comments to the Author**

1. Is the manuscript technically sound, and do the data support the conclusions?

Reviewer #1: Yes

Reviewer #2: Yes

2. Has the statistical analysis been performed appropriately and rigorously? 

Reviewer #1: I Don't Know

Reviewer #2: Yes

3. Have the authors made all data underlying the findings in their manuscript fully available?

Reviewer #1: Yes

Reviewer #2: Yes

4. Is the manuscript presented in an intelligible fashion and written in standard English?

Reviewer #1: Yes

Reviewer #2: Yes

5. Review Comments to the Author

Reviewer #1: This observational paper describes some interesting phenotypes in hMSCs under inflammatory conditions, including the upregulation of CD40 by inflammatory cytokines or ARDS edema fluid. Given the potential for hMSC treatment in ARDS patients, these important findings could lead to further investigations of the regulation of hMSC function and efficacy in ARDS. However, given that the current data does not comprehensively describe any novel mechanistic regulation of hMSC function by each of the different inflammatory stimuli, the authors could include some further studies/explanations to further enhance the insight provided by their work.

MAJOR

1. Which cell markers were used to define the hMSCs? Given that the cells were obtained from 3 different locations/sources, how did the authors ensure that the cells were consistent?

2. Regarding the different cytokine combinations, text in the introduction states that secretion of IL1b, TNFa, IL6, and IL8 is increased. Later in the introduction, the cytomix of TNFa, IFNy, and IL1b is mentioned. In the methods, the cytomix is described and another stimulus is introduced, i.e. LPS. Do the concentrations of cytokines and LPS used compare similarly to the circulating levels found in ARDS patient samples? Data describing these levels would be useful. Alternatively, authors could provide a clear rationale describing why the respective cytokines and concentrations were used.

3. Similarly, were the cytokine levels measured in the hydrostatic and ARDS edema fluids? These levels could be provided for comparison, along with suggestions for future studies that would help elucidate the different mechanisms responsible for how/why ARDS but not hydrostatic edema fluid resulted in increased CD40 expression.

4. In terms of the potential for treatment of ARDS using hMSCs, and given the pluripotency demonstrated, which cell phenotype do the authors expect to be most beneficial? It may be useful to the reader if the authors could add data or an explanation describing how this immunomodulatory phenotype could be maintained in clinical studies, especially given that the cytokine levels in human ARDS will likely vary over time and between individuals at different stages of disease progression.

5. It would be useful to quantify the findings in Figure 3.

MINOR

6. A few sentences in the Introduction would benefit from references being added. For example, “Alveolar macrophage activation leads to secretion of inflammatory cytokines, including interleukin-1β (IL-1�), tumor necrosis factor (TNF), interleukin-6 (IL-6), and interleukin-8 (IL-8), which lead to additional neutrophil activation.”, “hMSCs have many beneficial immunomodulatory effects.”, and “Through soluble mediators such as indoleamine-2,3-dioxygenase and PGE2, hMSCs promote the M1 to M2 transition of activated macrophages.”

7. Regarding the hydrostatic pulmonary edema fluid, it may not be clear to the reader from whom this fluid was taken. For example, what were the demographics of these individuals/patients?

8. For consistency, the heading “Flow cytometry – “ in the Methods could be replaced with “Flow cytometry.”, and “CD40 is expressed on the cell surface after CytoMix Exposure” replaced with “CD40 is expressed on the cell surface after CytoMix exposure”.

9. For clarity, regarding the following sentence in the Discussion, it may be useful to expand this sentence by including an explanation of the type of methods/studies that would be used to investigate/induce preconditioning: “Given the robust nature of this phenotype under specific conditions and the variability among human disease states, preconditioning may be important to ensure a uniform hMSC exposure in clinical trials.”

10. Given the observational nature of the findings, it may be more suitable and reflective of the findings to remove the following phrase from the title: ” – a mechanism for immune modulation”

11. According to the Methods, Mann Whitney or T-tests were used throughout. It is my understanding, however, that ANOVA should be used for multiple group comparisons (i.e. when there are more than 2 groups being compared).

Reviewer #2: This is an interesting paper by Wilfong et al that shows upregulated CD40 on human mesenchymal stem/stromal cells (hMSCs) after exposure to proinflammatory cytokines or ARDS pulmonary edema fluid. Furthermore, the expression of several anti-inflammatory cytokines and paracrine effectors, on hMSCs was also up-regulated. Finally, proinflammatory cytokines have been shown to reduce pleuripotency of hMSCs and block hMSCs differentiation into adipocytes. Overall, this is a very well written and a complete study on the important role of hMSCs in immunomodulation in ARDS. I only have a few minor comments that will help to clarify a few points in the manuscript.

Minor comments:

1. In Figure 2 and 3, quantification is required.

2. Please check and define all abbreviations on the first occurrence in context.

3. In the Statistical Analysis section, The description “P < 0.05 was considered significant” is missing.

6. PLOS authors have the option to publish the peer review history of their article (what does this mean?). If published, this will include your full peer review and any attached files.

Reviewer #1: **Yes: **Colin E. Evans

Reviewer #2: No

---

## [Author Response · Author response to Decision Letter 0]

21 Sep 2020

Editorial Office:

This has been done.

We apologize for this oversight. This data has been added as a supplemental figure. We have also expanded the statistical analysis section and added a new S3 data appendix to improve data transparency in accordance with your statistical reporting guidelines.

Reviewer #1:

This observational paper describes some interesting phenotypes in hMSCs under inflammatory conditions, including the upregulation of CD40 by inflammatory cytokines or ARDS edema fluid. Given the potential for hMSC treatment in ARDS patients, these important findings could lead to further investigations of the regulation of hMSC function and efficacy in ARDS. However, given that the current data does not comprehensively describe any novel mechanistic regulation of hMSC function by each of the different inflammatory stimuli, the authors could include some further studies/explanations to further enhance the insight provided by their work.

MAJOR

1. Which cell markers were used to define the hMSCs? Given that the cells were obtained from 3 different locations/sources, how did the authors ensure that the cells were consistent?

We felt that using hMSCs from multiple sources demonstrated that the phenotype described was consistent and robust for , increasingly the likelihood of reproducibility if studies were to be repeated in other laboratories. Additionally, this avoided the problem of a finding that might be reflective of cells from a specific location/source, rather than broadly applicable to hMSCs. All cell lines were validated using qPCR for the presence of CD73, CD90, and CD105, as well as the absence of CD11b, CD14, CD34, CD45, CD19, CD79A, CD54, and HLADRB. This data is shown in supplemental figure S1. This has been added to the methods. (Lines 80-82)

2. Regarding the different cytokine combinations, text in the introduction states that secretion of IL1b, TNFa, IL6, and IL8 is increased. Later in the introduction, the cytomix of TNFa, IFNy, and IL1b is mentioned. In the methods, the cytomix is described and another stimulus is introduced, i.e. LPS. Do the concentrations of cytokines and LPS used compare similarly to the circulating levels found in ARDS patient samples? Data describing these levels would be useful. Alternatively, authors could provide a clear rationale describing why the respective cytokines and concentrations were used.

The CytoMix cytokine cocktail was selected as it has been shown to recapitulate the effects of acute lung injury pulmonary edema fluid on alveolar type II cells. This sentence has been added to the introduction. (Lines 66-69)

3. Similarly, were the cytokine levels measured in the hydrostatic and ARDS edema fluids? These levels could be provided for comparison, along with suggestions for future studies that would help elucidate the different mechanisms responsible for how/why ARDS but not hydrostatic edema fluid resulted in increased CD40 expression.

We agree that measure the cytoking levels in hydrostatic and ARDS edema fluids could be information. Unfortunately, due to very limited sample availability, cytokine levels from these pulmonary edema fluid samples were not measured. 

4. In terms of the potential for treatment of ARDS using hMSCs, and given the pluripotency demonstrated, which cell phenotype do the authors expect to be most beneficial? It may be useful to the reader if the authors could add data or an explanation describing how this immunomodulatory phenotype could be maintained in clinical studies, especially given that the cytokine levels in human ARDS will likely vary over time and between individuals at different stages of disease progression.

Thank you for this recommendation. We have modified the discussion to say, “Given the robust nature of this phenotype under specific conditions and the variability among human disease states, preconditioning with CytoMix to induce an immunomodulatory phenotype may be important to ensure a uniform hMSC exposure in clinical trials, but this will require additional in vivo investigations.” (Lines 282-284) At present, we have not conducted any in vivo clinical trials to look at these immunomodulatory phenotypes after administration.

5. It would be useful to quantify the findings in Figure 3.

Thank you for this suggestion. These have been quantified within the Figure 3 legend, within the text, and in the S3 data appendix.

MINOR

6. A few sentences in the Introduction would benefit from references being added. For example, “Alveolar macrophage activation leads to secretion of inflammatory cytokines, including interleukin-1β (IL-1�), tumor necrosis factor (TNF), interleukin-6 (IL-6), and interleukin-8 (IL-8), which lead to additional neutrophil activation.”, “hMSCs have many beneficial immunomodulatory effects.”, and “Through soluble mediators such as indoleamine-2,3-dioxygenase and PGE2, hMSCs promote the M1 to M2 transition of activated macrophages.”

Thank you for this suggestion. Additional references have been added. The sentence stating ‘hMSCs have many beneficial immunomodulatory effects” is an introductory sentence to summarize the remainder of that paragraph that delineates these benefits. 

7. Regarding the hydrostatic pulmonary edema fluid, it may not be clear to the reader from whom this fluid was taken. For example, what were the demographics of these individuals/patients?

While this would indeed be interesting, this information is not available.

8. For consistency, the heading “Flow cytometry – “ in the Methods could be replaced with “Flow cytometry.”, and “CD40 is expressed on the cell surface after CytoMix Exposure” replaced with “CD40 is expressed on the cell surface after CytoMix exposure”.

These changes have been made.

9. For clarity, regarding the following sentence in the Discussion, it may be useful to expand this sentence by including an explanation of the type of methods/studies that would be used to investigate/induce preconditioning: “Given the robust nature of this phenotype under specific conditions and the variability among human disease states, preconditioning may be important to ensure a uniform hMSC exposure in clinical trials.”

We have added that additional in vivo experiments would be required. We respectfully feel that a larger discussion would take away from the main finding of this report; namely the finding of CD40 positivity.

10. Given the observational nature of the findings, it may be more suitable and reflective of the findings to remove the following phrase from the title: ” – a mechanism for immune modulation”

We have added the word “potential” before mechanism, as these are observational studies and direct causation is not established.

11. According to the Methods, Mann Whitney or T-tests were used throughout. It is my understanding, however, that ANOVA should be used for multiple group comparisons (i.e. when there are more than 2 groups being compared).

ANOVA, or Kruskal-Wallis-given our data is non-parametric, evaluate if there are differences between any of the groups in a multiple comparison analysis, but cannot evaluate which groups were different. For the second portion of the analysis, we used Mann-Whitney U tests to ascertain between group comparisons. The Kruskal-Wallis ANOVA was <0.01 for panels B-F of Figure 2. This has been added to the caption. 

Reviewer #2:

This is an interesting paper by Wilfong et al that shows upregulated CD40 on human mesenchymal stem/stromal cells (hMSCs) after exposure to proinflammatory cytokines or ARDS pulmonary edema fluid. Furthermore, the expression of several anti-inflammatory cytokines and paracrine effectors, on hMSCs was also up-regulated. Finally, proinflammatory cytokines have been shown to reduce pleuripotency of hMSCs and block hMSCs differentiation into adipocytes. Overall, this is a very well written and a complete study on the important role of hMSCs in immunomodulation in ARDS. I only have a few minor comments that will help to clarify a few points in the manuscript.

Minor comments:

1. In Figure 2 and 3, quantification is required.

We have now performed quantification for both Figure 2 and Figure 3. 

2. Please check and define all abbreviations on the first occurrence in context.

We have reviewed abbreviations, and made updates as necessary to define upon first occurrence in the text.

3. In the Statistical Analysis section, The description “P < 0.05 was considered significant” is missing.

Thank you for noting this. We have expanded the statistical analysis section and included this.

---

## [Decision Letter · Decision Letter 1]

24 Sep 2020

Proinflammatory cytokines and ARDS pulmonary edema fluid induce CD40 on human mesenchymal stromal cells – a potential mechanism for immune modulation

PONE-D-20-17990R1

Dear Dr. Wilfong,

We’re pleased to inform you that your manuscript has been judged scientifically suitable for publication and will be formally accepted for publication once it meets all outstanding technical requirements.

Kind regards,

You-Yang Zhao

Academic Editor

PLOS ONE

Additional Editor Comments (optional):

Reviewers' comments:

Reviewer's Responses to Questions

**Comments to the Author**

1. If the authors have adequately addressed your comments raised in a previous round of review and you feel that this manuscript is now acceptable for publication, you may indicate that here to bypass the “Comments to the Author” section, enter your conflict of interest statement in the “Confidential to Editor” section, and submit your "Accept" recommendation.

Reviewer #1: All comments have been addressed

Reviewer #2: All comments have been addressed

2. Is the manuscript technically sound, and do the data support the conclusions?

Reviewer #1: Yes

Reviewer #2: Yes

3. Has the statistical analysis been performed appropriately and rigorously? 

Reviewer #1: Yes

Reviewer #2: Yes

4. Have the authors made all data underlying the findings in their manuscript fully available?

Reviewer #1: Yes

Reviewer #2: Yes

5. Is the manuscript presented in an intelligible fashion and written in standard English?

Reviewer #1: Yes

Reviewer #2: Yes

6. Review Comments to the Author

Reviewer #1: (No Response)

Reviewer #2: The authors have addressed the detailed questions I raised in my previous review. I have no further comments regarding this manuscript.

7. PLOS authors have the option to publish the peer review history of their article (what does this mean?). If published, this will include your full peer review and any attached files.

Reviewer #1: **Yes: **Colin Evans

Reviewer #2: No

---

## [Editor Report · Acceptance letter]

28 Sep 2020

PONE-D-20-17990R1 

Proinflammatory cytokines and ARDS pulmonary edema fluid induce CD40 on human mesenchymal stromal cells – a potential mechanism for immune modulation 

Dear Dr. Wilfong:

I'm pleased to inform you that your manuscript has been deemed suitable for publication in PLOS ONE. Congratulations! Your manuscript is now with our production department. 

Kind regards, 

on behalf of

Dr. You-Yang Zhao 

Academic Editor

PLOS ONE